# Changes in Bone Metabolism in Patients with Rheumatoid Arthritis during Tumor Necrosis Factor Inhibitor Therapy

**DOI:** 10.3390/jcm12051901

**Published:** 2023-02-28

**Authors:** Tanja Janković, Momir Mikov, Jelena Zvekić Svorcan, Ivana Minaković, Jelena Mikov, Ksenija Bošković, Darko Mikić

**Affiliations:** 1Medical Faculty, University of Novi Sad, Special Hospital for Rheumatic Diseases, 21000 Novi Sad, Serbia; 2Department of Pharmacology and Toxicology, Medical Faculty, University of Novi Sad, 21000 Novi Sad, Serbia; 3Medical Faculty, University of Novi Sad, Health Center “Novi Sad”, 21000 Novi Sad, Serbia; 4Medical Faculty, University of Novi Sad, 21000 Novi Sad, Serbia; 5Pathology and Forensic Medicine Institute, Medical Faculty of the Military Medical Academy, University of Defence, 11000 Belgrade, Serbia

**Keywords:** rheumatoid arthritis, bone metabolism, bone mineral density, osteoporosis inhibitor tumor necrosis factor alpha

## Abstract

Tumor necrosis factor alpha (TNF-α), which enhances osteoclast activity and bone resorption, is one of the key inflammation mediators in rheumatoid arthritis (RA). The aim of this study was to assess the influence of yearlong TNF-α inhibitor application on bone metabolism. The study sample comprised 50 female patients with RA. Analyses involved the osteodensitometry measurements obtained using a “Lunar” type apparatus and the following biochemical markers from serum: procollagen type 1 N-terminal propeptide (P1NP), beta crosslaps C-terminal telopeptide of collagen type I (b-CTX) by ECLIA method, total and ionized calcium, phosphorus, alkaline phosphatase, parathyroid hormone and vitamin D. Analyses revealed changes in bone mineral density (BMD) at L1–L4 and the femoral neck, with the difference in mean BMD (g/cm^2^) not exceeding the threshold of statistical significance (*p* = 0.180; *p* = 0.502). Upon completion of 12-month therapy, a significant increase (*p* < 0.001) in P1NP was observed relative to b-CTX, with mean total calcium and phosphorus values following a decreasing trend, while vitamin D levels increased. These results suggest that yearlong application of TNF inhibitors has the capacity to positively impact bone metabolism, as indicated by an increase in bone-forming markers and relatively stable BMD (g/cm^2^).

## 1. Introduction

Rheumatoid arthritis (RA) is a systemic, autoimmune, chronic inflammatory disease that affects bone metabolism by increasing bone resorption, leading to osteoporosis and a high risk of fracture [1,2]. The presence of numerous proinflammatory cytokines in RA is associated with the activation of osteoclastogenesis, with tumor necrosis factor alpha (TNF-α) playing a leading role in this process. TNF-α stimulates the expression of the receptor activator of nuclear factor kB ligand (NF-kB), RANKL, which is instrumental in osteoclast differentiation and maturation. It can also exert its influence through the soluble receptor osteoprotegerin (OPG), or via stromal cells of the bone marrow lineage of osteoblasts, as well as by directly activating the cells in osteoclast lineage [3]. The interaction between RANKL and its receptor-activator of nuclear factor kB (RANK) occurs on the surface of osteoclasts by forming a RANKL/RANK bond that leads to accelerated differentiation, maturation, activation and life extension of osteoclasts, which initiates the bone resorption process. The OPG soluble part of the receptor for RANKL, which is structurally homologous to RANK, by binding to RANKL forms a RANKL/OPG bond that inhibits the final phase of osteoclast differentiation as well as activation of matrix osteoclast suppression, while accelerating osteoclast apoptosis. Therefore, balance between RANKL and OPG is the main regulatory factor of the biological balance involving bone formation and resorption [4,5,6]. TNF-α can stimulate osteoclast precursors directly through TNF receptor 1 (TNFR1) signaling, while soluble TNF is responsible for mobilization of osteoclasts from the bone marrow. Binding of TNF-α to receptors activates intracellular cascades that include NF-κB and mitogenic activation of protein kinase, thus transmitting information from the receptor to the nucleus [7]. Osteoblast differentiation and activity are influenced by TNF-α, while NF-kB signaling inhibits the regulation of bone morphogenetic protein (BMP) [8]. As Wnt uses a co-receptor (frizzled receptor and lipoprotein receptor 5 − LPR5) to activate, its transduction pathway is presently considered the most important for the control of osteoblast differentiation. In addition, in interaction with the Wnt signaling pathway, TNF-α plays an important role in the control and differentiation of osteoblasts, as TNF-α stimulates Dkk-1 (Dickkopf-related protein 1) expression of the endogenous Wnt signal inhibitor. During this process, Dkk-1 binds to the LDL-receptor LRP5 or LRP6 of osteoblasts, which inhibits their activity and enhances osteoclastogenesis [9]. By stimulating the production of sclerostin—a hormone produced by osteocytes—TNF reduces preosteoblast differentiation into osteoblasts and thus inhibits bone tissue formation [10,11].

Clinical tracking of the bone remodeling process consists of the determination of biochemical markers of bone resorption and bone formation in serum and urine, allowing bone metabolism, bone resorption rate and fracture risk to be assessed [12,13]. Together with the determination of inorganic matrix markers (i.e., total and ionized calcium, phosphorus and alkaline phosphatase), hormones closely related to bone metabolism—parathyroid hormone (PTH) and vitamin D—are also determined. The International Osteoporosis Foundation (IOF) and International Federation of Clinical Chemistry and Laboratory Medicine (IFCC) recommend tracking of biochemical marker serum procollagen type I N propeptide (P1NP) as an indicator of bone synthesis and Beta-CrossLaps/serum C-terminal cross-linking telopeptide of type I collagen (b-CTX) as an indicator of bone resorption [14,15]. According to the European League Against Rheumatism (EULAR) guidelines, introduction of biological drugs is recommended in patients with highly active RA that have not responded well to the previously applied therapy [16].

The application of biological drugs—various proteins with immunomodulation capability (i.e., direct regulation of immune response)—has initiated a new concept in RA treatment. Some of the most frequently applied biological drugs are so-called TNF inhibitors, as their mechanism of action is based on blocking proinflammatory cytokine TNF-α. Besides their desirable influence on disease activity, these drugs impact local and general loss of bone mineral density (BMD). Presently, different TNF inhibitors are utilized in RA treatment, including etanercept, adalimumab, golimumab, infliksimab and certolizumab pegol [17,18]. However, TNF inhibitor choice should be individualized to each patient. Etanercept is applied as subcutaneous injections, which are administered once a week in 50 mg doses, while both adalimumab (in 40 mg doses) and certolizumab pegol (in 50 mg doses) are administered at biweekly intervals, and golimumab is applied once a month in 50 mg doses. Infliksimab is applied intravenously starting with 3 mg/kg doses [19,20]. These medications can be particularly beneficial for mitigating the high rate of general bone mass loss in RA patients, given the evidence of 2.1–2.7% BMD loss in lumbar spine and 1.7–3.6% in femoral neck in this population [21]. Moreover, in findings yielded by the multi-center study conducted by Chopin et al., as a part of which the impact of 12-month infliksimab therapy on bone metabolism was tracked, this TNF inhibitor had the capacity to increase bone-forming markers and decrease bone resorption markers [22]. Similar results were obtained by Orsolini and colleagues based on treating 68 RA patients with infliksimab [23]. Their findings revealed a significant increase in the bone-forming marker P1NP accompanied by a significant decrease in the bone resorption marker beta-crosslaps.

As extant research mostly focused on the capacity of infliximab and adalimumab to induce changes in bone mineral density and the biochemical markers of bone resorption and synthesis, the aim of the present study was to evaluate the effectiveness of 12-month TNF-α inhibitor (etanercept, adalimumab, golimumab and infliximab) therapy in terms of bone metabolism improvements in patients with rheumatoid arthritis.

## 2. Materials and Methods

### 2.1. Study Design and Participants

This research involved 50 female patients in whom the rheumatoid arthritis diagnosis was made on the basis of American College of Rheumatology (ACR)/European League Against Rheumatism (EULAR) criteria published in 2010 [24], and whose clinical status indicated that they would benefit from biological drugs from the TNF inhibitor group. The study was conducted at the Special Hospital for Rheumatic Diseases Novi Sad, Serbia.

It was designed as a 12-month-long retrospective/prospective study that would include all patients treated with a biological drug from the TNF-inhibitor group in the Special Hospital for Rheumatic Diseases in Novi Sad, Serbia. In order to obtain valid results, our sample was as homogeneous as possible in terms of disease type, gender, age, disease length and activity, and treatment method. Our cohort comprised 50 female patients, which was sufficient, as 48 individuals were the minimum required to attain 10% margin of error and 85% confidence level.

The study inclusion criteria (all of which had to be present in an individual to be considered for participation in the study) were: menopause duration <5 years, RA diagnosis within the last 10 years, methotrexate (MTX) therapy (at a stable dose of 10 mg/week at a minimum), no glucocorticoid therapy, anatomical stage II of joint destruction, and Disease Activity Score-28 DAS28 > 5.1.

The study exclusion criteria (presence of any of which would preclude participation in the study) were: menopause duration > 5 years; RA duration > 10 years, anatomical stage III (destruction of bone as well as cartilage, joint deformities) or stage IV (fibrous or bony ankylosis, advanced muscle atrophy, joint deformities) of joint destruction; presence of any disease with adverse repercussions for bone tissue, such as malignant tumors, kidney and liver insufficiency, endocrine diseases (Cushing syndrome, hyperthyroidism, hyperparathyroidism, hypoparathyroidism, hyperprolactinemia, acromegaly); therapy involving drugs that impact bone metabolism (low-molecular-weight heparin, aromatase inhibitors, anticonvulsants, antipsychotics, antidiabetics, L-thyroxin in supraphysiological doses); and prolonged immobility.

All patients that were eligible for participation and were willing to take part in the study signed the informed consent form. Moreover, all participants received vitamin D supplements (800 IU per day) during the study period.

All performed procedures were in conformity with the ethical standards of the institutional research committee and with the Helsinki Declaration and its subsequent amendments or comparative ethical standards. The study was approved by the Ethics Committee of the Special Hospital for Rheumatic Diseases, Novi Sad, Serbia, protocol code 14/35-5/1-016, and by the Ethics Committee of the Medical Faculty, University of Novi Sad, Serbia, protocol code 01-39/148/1.

### 2.2. Measurements

At entry, all medical documentation was reviewed, rheumatologic examination was conducted, pain threshold was determined, and global RA activity was assessed through visual analogue scale (VAS). The VAS is a subjective measure of symptom severity; it was used in the present study to allow respondents to rate perceived extent of pain on a scale from 0 (indicating absence of pain) to 100 mm (corresponding to almost unbearable pain) [25].

For disease activity assessment, clinical index Disease Activity Score-28 with serum CRP levels (DAS28 CRP) and erythrocyte sedimentation rate (ESR) (DAS28 ESR) were adopted. The Disease Activity Score-28 is a combined index that measures disease activity in patients with rheumatoid arthritis based on both 28 tender joint count (TJC) and swollen joint count (SJC), as well as a laboratory measure of acute inflammation (serum CRP levels or erythrocyte sedimentation rate) and patient global health assessment (PGA) of disease severity on a 0–10 cm scale. The activity score can be calculated according to the following formula: DAS28-CRP = 0.56 * √(TJC28) + 0.28 * √(SJC28) + 0.36 * ln (CRP + 1) + 0.014 * (PGA) + 0.96; DAS28-ESR = 0.56 * √(TJC28) + 0.28 * √(SJC28) + 0.70 * ln (ESR) + 0.014 * (PGA) [26]. The final score of DAS28-CRP or ESR ≤ 2 indicates that the patient is in remission, while scores in the 2.6–3.2 and 3.2–5.1 ranges are indicative of low and moderate disease activity, respectively, and those above >5.1 signify high activity [26].

In addition to the aforementioned evaluations, a health assessment questionnaire (HAQ) was administered to all study participants to assess their degree of incapability. It comprises 20 items classified under eight categories pertaining to activities of daily living: dressing, arising, eating, walking, hygiene, reach, grip, and common daily activities. Each item is rated on a 0–3 scale, where 0 indicates “without difficulty” and 3 indicates “unable to do”, and additional points can be added if aids or devices are needed for performing specific activities. The final score is calculated by summing the scores obtained for each of the categories and dividing this value by the number of categories, resulting in the 0–3 range, where a higher figure indicates poorer quality of life [27].

The body mass index (BMI) is a quantitative ratio of body mass expressed in kilograms (kg) and body height expressed in meters squared (m^2^), and was used to classify the study participants into six groups: severely underweight (BMI ≤ 16.49 kg/m^2^), underweight (BMI = 16.50–18.49 kg/m^2^), normal weight (BMI = 18.50–24.99 kg/m^2^), overweight (BMI = 25.00–29.99 kg/m^2^), obese (BMI = 30.00–34.99 kg/m^2^), and severely obese (BMI ≥ 35.00 kg/m^2^) [28].

Bone mineral density (BMD) was measured in two regions—front-end lumbar spine (LS) region L1–L4 and the left proximal femoral neck—at the beginning and end of the 12-month TNF inhibitor treatment. BMD measurements were performed by the same technician using the dual energy X-ray absorptiometry (DXA) method on the “Lunar” type apparatus. The coefficient of variation (CV) in the measurements performed at our hospital was 0.8%, as determined daily using the anatomical spine phantom, and no machine drift was detected during the study period. The short-term in vivo precision error for L2–L4 lumbar spine is 0.012 g/cm^2^ (LSC = 0.034 g/cm^2^ at the 95% confidence level) and is 0.013 g/cm^2^ for femur neck (LSC = 0.035 g/cm^2^ at 95% confidence level). The obtained T-scores were used to classify the participants into three groups: normal (T-score > −1), osteopenia (T-score in the −1 to −2.5 range) and osteoporosis (T-score < −2.5) [29].

### 2.3. Biochemical Analysis

Blood tests were also performed at the beginning and end of the study, focusing on sedimentation (SE), C-reactive protein (CRP), ionized and total calcium, phosphorus, 25(OH) vitamin D, alkaline phosphatase, parathyroid hormone (PTH), and biochemical bone markers procollagen type I N propeptide (P1NP) and Beta-CrossLaps/serum C-terminal cross-linking telopeptide of type I collagen (b-CTX). Erythrocyte sedimentation rate was determined by the Westergren method, while ionized calcium and phosphorus were measured via the usual spectrophotometric laboratory procedure based on ion exchange using ion exchange electrodes. Alkaline phosphatase was measured by spectrophotometry, and P1NP and b-CTX were determined using ECLIA methods. Blood samples were collected between 7:00 a.m. and 9:00 a.m. after an overnight fast.

### 2.4. Statistical Analysis

All statistical analyses were performed using Windows SPSS ver. 24 (Statistical Package for the Social Sciences) with *p* ≤ 0.05 signifying statistical significance. First, the distribution of numerical variables was examined using the Kolmogorov–Smirnov test. As only age was found to be normally distributed, it was presented as mean ± SD (standard deviation), while median (IQR) was calculated in all other cases, and frequencies and percentages were reported for categorical variables. The differences between the parameters measured at two time points were tested by the non-parametric Wilcoxon signed-rank test (Z).

## 3. Results

### 3.1. Baseline Clinical Characteristics

The study sample comprised 50 female RA patients aged 51.50 ± 3.94 years, 84% of whom were diagnosed within the preceding 5 years, and 95% and 92% were positive for rheumatoid factor (RF) and anti-cyclic citrulline peptide antibodies (ACPA), respectively. All patients received methotrexate in stabile doses (15–17.5 mg/week) and were treated by TNF inhibitors as the first biological drug, whereby 46%, 34%, 18%, and 2% of the sample received adalimumab, etanercept, golimumab, and infliximab, respectively. Based on their BMI, 62% of the patients were mildly (either slightly or moderately) obese, and the remaining 36% and 2% had optimal weight and were severely obese, respectively (Table 1). TNF inhibitor therapy resulted in a statistically significant improvement in all disease activity parameters compared to the baseline. Specifically, DAS28 SE (Z = −5.71, *p* ˂ 0.001), SE (mm/h) (Z = −5.97, *p* ˂ 0.001) and CRP (Z = −5.90, *p* ˂ 0.001) exhibited statistically significant decreases.

### 3.2. Effect of TNF Inhibitors on Bone Mineral Density

The obtained findings indicated changes in the measured BMD (g/cm^2^) values after 12-month TNF inhibitor therapy relative to the baseline, but the difference did not exceed the threshold of statistical significance (Z = 1.34, *p* = 0.180; Z = 0.67, *p* = 0.502), as shown in (Table 2).

Prior to initiating the TNF inhibitor therapy, all but one patient had P1NP values in the reference range (16.3–73.9 ng/mL). Upon therapy completion, all patients had P1NP values in the reference range, and the overall improvement in this biomarker (42.30 [IQR = 21.67] vs. 59.30 [IQR = 18.27]) was statistically significant (Z = −6.07, *p* ˂ 0.001). Although a majority of patients (78%) had the biochemical marker b-CTX in serum values within the reference range (556–1008 ng/mL) before starting the TNF inhibitor therapy, upon its completion, this percentage increased to 94%, (593.00 [IQR = 63.00] vs. 627.50 [IQR = 100.00]), and the overall increase in this parameter was statistically significant (Z = −4.78, *p* ˂ 0.001). Analyses further revealed that BMI impacted changes in both P1NP and b-CTX values, whereby the greatest increases in P1NP and b-CTX were observed in patients with optimal body mass (Table 3).

A further goal of our analysis was to determine if BMD, P1NP and b-CTX changes related to different TNF inhibitors were statistically significant. Since only 2% of patients received infliximab, its influence on the observed parameters was not assessed. Moreover, changes induced by the remaining three TNF inhibitors in T-score (SD) and BMD (g/cm^2^) were comparable. Even though the differences were not statically significant, the greatest increase in P1NP was noted in patients treated with golimumab, while those treated with etanercept had the greatest increase in b-CTX (Table 4).

### 3.3. Changes in Other Parameters of Bone Metabolism after One-Year Use of TNF Inhibitors

The values of other observed bone metabolism parameters also changed following the 12-month TNF inhibitor therapy. Specifically, a decrease in average values was noted for total calcium (from 2.30 [IQR = 0.24] to 2.30 [IQR = 0.20]; Z = −3.07, *p* = 0.002), ionized calcium (from 1.12 [IQR = 0.09] to 1.10 [IQR = 0.06]; Z = −4.35, *p* ˂ 0.001), and phosphorus (from 1.00 [IQR = 0.13] to 1.00 [IQR = 0.10]; Z = −2.55, *p* = 0.011). While 25(OH) vitamin levels increased from 44.00 [IQR = 20.00] mol/L to 51.50 [IQR = 16.25] (Z = −5.06, *p* ˂ 0.001), no statistically significant changes were noted in the average values of alkaline phosphatase (Z = −1.53, *p* = 0.124).

Two-way analysis of variance (ANOVA) was also conducted to examine the joint effect of TNF-α inhibitors and vitamin D on the examined bone metabolism parameters. The obtained results did not exceed the threshold of statistical significance (*p* = 0.037).

## 4. Discussion

In the extant literature, the presence of numerous proinflammatory cytokines in rheumatoid arthritis (RA) is associated with localized inflammatory bone resorption and generalized bone loss. Available findings further indicate that the receptor activator of nuclear factor kB (RANK)-RANK ligand (RANKL) system is the main driver of inflammatory bone resorption [30]. On the other hand, the use of TNF inhibitors has been shown to affect bone metabolism by increasing the P1NP and osteocalcin (OC) serum level (as bone synthesis markers), and decreasing the levels of serum b-CTX and RANKL (as bone resorption markers in RA), thus slowing down generalized osteoporosis and the development of periarticular erosions [31].

Guided by this evidence, the aim of the present study was to evaluate the impact of 12-month use of TNF inhibitors on changes in bone mineral density and biochemical markers of bone synthesis (P1NP) and resorption (b-CTX). These markers were chosen, as our sample comprised solely women aged 51.50 ± 3.94 years, and it is widely known that estrogen affects bone homeostasis directly (via the effect on bone cells through the RANKL-RANK-OPG pathway), as well as indirectly through an immune mechanism by mitigating the increase in cytokine (such as IL-1, IL-6 and TNF) production caused by estrogen deficiency. In this context, TNF-α (produced by T-lymphocytes in the bone marrow) is particularly significant, as it is the most influential cytokine in bone loss caused by estrogen deficiency. In our cohort, menopause duration did not exceed 5 years, which is relevant as estrogen levels decrease rapidly during menopause, which in turn activates the differentiation and proliferation of osteoclasts while inhibiting the action of osteoblasts and increasing osteocyte apoptosis. All of these effects accelerate bone resorption, as confirmed by extant evidence indicating that the onset of estrogen decline coincides with a phase of rapid bone loss, resulting in 10% and 5% bone mass loss in the spine and hip, respectively, during the 5 five years of menopause. Some estimates further suggest that after 10–15 years of menopause, about 50% of trabecular and 30% of cortical bone is lost [32]. Our findings indicate that 12-month use of TNF inhibitors attenuates the decline in BMD measured at the L1–L4 level as well as in the femoral neck area. Moreover, although the increase in BMD values for the lumbar spine and hip did not exceed the threshold of statistical significance, we observed statistically significant improvements in the corresponding T-score (SD) values. Our findings are in agreement with the results reported by Nutz et al. based on a meta-analysis of 15 studies that included 12-month TNF-inhibitor treatment in patients with RA [33].

Similarly, Zerbini and colleagues analyzed 28 studies in which participants received TNF inhibitors and observed their beneficial effect on preserving or increasing BMD in the lumbar spine and hip, as well as achieving a better biochemical bone marker profile [34]. Findings reported by other authors yielded similar conclusions [35,36,37,38]. It is also worth noting that, by analyzing the effect of TNF inhibitors on bone metabolism in RA patients over a 15-month period, Jura-Poltorak et al. confirmed that they are effective in arresting bone loss. However, in their cohort, the improvements in BMD in the lumbar spine and femoral neck, as well as the corresponding T scores, failed to reach statistical significance. On the other hand, as the authors observed a statistically significant change in the levels of bone synthesis/resorption biochemical markers, it appears that these markers respond more rapidly to TNF inhibitor therapy compared to bone mineral density [39].

In the present study, a higher degree of bone resorption was associated with greater disease activity and b-CTX levels, which correlated with lower BMD values at the onset of TNF inhibitor therapy. Therefore, by monitoring b-CTX levels, we could predict further BMD loss. Our analyses also revealed that TNF inhibitors increased serum P1NP levels and mitigated the b-CTX increase, thus elevating the P1NP/b-CTX ratio. Moreover, TNF inhibitor use was positively correlated with changes in bone biochemical marker levels, which indicated a statistically significant increase in P1NP levels as well as improvements in b-CTX (which did not exceed the threshold of statistical significance). Finally, the observed changes in P1NP and b-CTX upon completion of 12-month TNF inhibitor administration were associated with RA disease activity reduction. Similar findings were reported by Szulc and colleagues based on an investigation involving 54 patients treated with TNF inhibitors for 12 months [40]. However, other authors reported statistically significant improvements in both parameters, while some also noted BMD increases in their cohorts [39].

To position our results in this context, we segregated our sample into younger (below 50 years) and older (50+ years) groups and examined their outcomes. These comparisons revealed that greater improvements in P1NP and b-CTX were associated with younger age (30% and 10.5% for those under 50 vs. 26.8% and 6.6% for women over 50). Likewise, when the cohort was segregated by BMI, we determined that being underweight (BMI < 19), as well as overweight or obese (BMI > 25), was a risk factor for osteoporosis, as it resulted in more pronounced changes in bone metabolism. These observations concur with the available data, indicating that BMD declines more rapidly in obese individuals due to inadequate physical activity, hypertension, and suboptimal vitamin D levels [41]. In addition, adipose tissue secretes cytokines that affect bone tissue by increasing bone resorption, while adipokines affect the central nervous system by altering the influence of the sympathetic nervous system on bone tissue [42]. Therefore, it is not surprising that the greatest changes in P1NP values (38%) and b-CTX (8.4%) were recorded in patients of normal weight.

Hypovitaminosis D is highly prevalent in patients suffering from inflammatory rheumatic diseases, especially RA, and may exacerbate the negative impact of inflammation on BMD. Vitamin D in its active form—25(OH)D—exhibits an immunoregulatory effect that manifests through the regulation of monocytes and macrophages as well as the activity of B and T cells. It is thus believed that vitamin D intake can reduce the production of proinflammatory cytokines such as TNF-α and IL-6, which play a key role in bone resorption in RA patients. Consequently, the potential beneficial role of vitamin D as a modulator of inflammation in RA is increasingly being discussed [43].

In our cohort, 12-month TNF inhibitor therapy resulted in a statistically significant reduction in disease activity (as measured by composite indices DAS28 SE and DAS28 CRP), while increasing average vitamin D, total calcium, ionized calcium, and phosphorus values. These observations are supported by the findings reported by Oelzner et al., indicating that high RA activity is associated with changes in vitamin D metabolism and increased bone resorption [44], which is expected given that lower serum vitamin D levels may contribute to negative calcium balance and inhibition of bone formation. Similar conclusions were reached by Scott et al. and Lin et al. based on their meta-analyses of available studies [45,46]. However, given the paucity of comparative studies focusing on the efficacy of different TNF inhibitors (mostly tocilizumab and abatacept), our findings regarding the changes in BMD and bone synthesis/resorption biochemical markers can only be directly compared with those reported by Sainaghi and colleagues, with which they concur [47]. It is also worth noting that, even though 12-month administration of all tested biological drugs resulted in an increase in P1NP and b-CTX, the highest percentage increase in P1NP was achieved with adalimumab (34%), followed by golimumab (32.6%) and finally etanercept (18.7%). In contrast, b-CTX was most significantly modified by etanercept (9.21%), followed by adalimumab (7.48%) and finally by golimumab (4.94%).

The significance of our study derives from the inclusion of biologically naïve patients who were selected based on clearly defined criteria, excluding individuals taking glucocorticoids and other drugs as well as those suffering from diseases that are known to impact bone remodeling (which would affect any changes in BMD that occurred during the study period). This rigorous research design allowed us to obtain a clearer picture of the effect of TNF inhibitors on BMD. Furthermore, several variables—SE, CRP, PTH, vitamin D 25(OH) level, DAS28 SE and DAS28 CRP, and self-reported quality of life—that could potentially affect bone metabolism were monitored. Based on these analyses, we hypothesize that TNF inhibitors are not only effective in controlling inflammation, but may also directly inhibit osteoclast activity.

Nonetheless, when interpreting our findings, several limitations to our study should be considered, one of which was a small sample size, which was partly due to the strict study inclusion/exclusion criteria, but was also a result of stringent approval protocols implemented for the administration of biological drugs (including TNF inhibitors). Therefore, as a part of our investigation, we were unable to detect minor changes in the observed variables. Moreover, the 12-month follow-up period was potentially insufficient for emergence of meaningful BMD improvements in our patient cohort. A further limitation of the present study stems from the lack of a control group that would have allowed us to better understand the effect of TNF inhibitors on bone marker values, as any changes in the treatment group could be compared with both the baseline and the controls. This shortcoming should be rectified in future studies, allowing the potential beneficial effects of biological drugs, including TNF inhibitors, on inflammatory bone loss to be assessed more precisely.

## 5. Conclusions

In our cohort comprising 50 female RA patients, 12-month TNF inhibitor therapy had a positive effect on bone metabolism. Increased P1NP and b-CTX values were accompanied by rapid bone remodeling, which was not dependent on the changes in BMD values. Moreover, TNF inhibitor application prevented a decline in BMD (g/cm^2^), whereby changes in BMD values did not differ among examined biological drugs.

## Figures and Tables

**Table 1 jcm-12-01901-t001:** Baseline clinical characteristics of rheumatoid arthritis patients.

Clinical Characteristics	RA (n = 50)
Female patients, n (%)	50 (100.0%)
Age (years), mean ± SD	51.50 ± 3.94
RA duration, n (%)	
1–5 years	42 (84.0%)
5–10 years	8 (16.0%)
Rheumatoid factor, n (%)	
Negative	13 (26.0%)
Positive	37 (74.0%)
ACPA, n (%)	
Negative	1 (2.0%)
Positive	49 (98.0%)
Biological drug, n (%)	
Etanercept	17 (34.0%)
Adalimumab	23 (46.0%)
Golimumab	9 (18.0%)
Infliximab	1 (2.0%)
BMI (kg/m^2^) category, n (%)	
Normal weight (18.50–24.99)	18 (36.0%)
Slightly overweight (25.00–29.99)	24 (48.0%)
Obese (30.00–34.99)	7 (14.0%)
Severely obese (≥35)	1 (2.0%)

n—number of patients; SD—standard deviation; RA—rheumatoid arthritis; ACPA anti-cyclic citrulline peptide antibodies; BMI—body mass index; kg—kilogram, m^2^—meter squared.

**Table 2 jcm-12-01901-t002:** Differences between the average values of T-score and BMD after 12-month TNF inhibitor therapy.

	N	Min	Max	Me	IQR	Z	*p*
BMD (g/cm^2^) L1–L4, first measurement	50	0.78	1.19	0.96	0.10	1.34	0.180
BMD (g/cm^2^) L1–L4, second measurement	50	0.75	1.20	0.97	0.12		
BMD (g/cm^2^) femoral neck, first measurement	50	0.64	1.09	0.84	0.11	0.67	0.502
BMD (g/cm^2^) femoral neck, second measurement	50	0.71	1.07	0.84	0.14		

N—number of patients; Min—minimum; Max—maximum; Me—median; IQR—interquartile range; Z—Wilcoxon signed-rank test; *p*—statistical significance (*p* ≤ 0.05).

**Table 3 jcm-12-01901-t003:** BMI and improvements in the P1NP and b-CTX values.

	BMI (18.5–24.9)			BMI (25+)		
	Me	IQR	% Change	Me	IQR	% Change
P1NP, first measurement	40.20	16.83		42.85	22.25	
P1NP, second measurement	59.30	12.78	47.51%	57.70	21.85	34.65%
b-CTX, first measurement	593.50	72.50		593.00	68.75	
b-CTX, second measurement	632.50	99.85	6.57%	626.00	104.50	5.56%

BMI—body mass index; Me—median; IQR—interquartile range; P1NP—procollagen type 1N-terminal propeptide; b-CTX—beta crosslaps C-terminal telopeptide of collagen type I.

**Table 4 jcm-12-01901-t004:** Percentage change in P1NP and b-CTX with respect to the prescribed TNF inhibitor.

	Biological Drug Treatment								
	Etanercept			Adalimumab			Golimumab		
	Me	IQR	% Change	Me	IQR	% Change	Me	IQR	% Change
P1NP, 1st measurement	44.50	25.80	34.38%	42.30	14.30	41.37%	38.40	25.78	53.12%
P1NP, 2nd measurement	59.80	27.05		59.80	15.43		58.80	20.15	
b-CTX 1st measurement	620.00	70.00	8.87%	589.00	71.00	6.09%	583.00	43.45	4.97%
b-CTX 2nd measurement	675.00	95.65		624.90	100.00		612.00	48.75	

Me—median; IQR—interquartile range; P1NP—procollagen type 1N-terminal propeptide; b-CTX—beta crosslaps C-terminal telopeptide of collagen type I.

## Data Availability

The data presented in this study are available on request from the corresponding author.

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
