# Peer review of "Changes in Bone Metabolism in Patients with Rheumatoid Arthritis during Tumor Necrosis Factor Inhibitor Therapy"

_jcm, 2023, doi:10.3390/jcm12051901_

Round 1

Reviewer 1 Report

Subject Appropriateness of the Manuscript:

The topic of this manuscript falls within the scope of JCM

Comments

In the manuscript entitled: “Changes in bone metabolism in patients with rheumatoid arthritis during tumor necrosis factor inhibitor therapy” the Authors aimed to investigate the effect of the one year therapy with TNF-a inhibitors on the bone metabolism assessed by biochemical reference markers of bone synthesis (P1NP - procollagen type 1N-terminal propeptide)  and bone resorption  (b-CTX - beta crosslaps C-terminal telopeptide of collagen type I beta-crosslaps) and inorganic matrix markers (total and ionized calcium, phosphorus and alkaline phosphatase) as well as hormones closely related to bone metabolism such as parathyroid hormone (PTH) and vitamin D level.  The effectiveness of 12-month therapy with TNF inhibitor (etanercept, adalimumab, golimumab and infliximab) on bone metabolism improvements in 50 female patients with rheumatoid arthritis was assessed in this study. Authors conclude that yearlong application of TNF inhibitors has the positive effect on bone metabolism, as indicated by an increase in bone-forming markers. The undertaken research topic is of significant importance in clinical practice.

The manuscript, however, has some limitations, the most important of which are:

1.     a small number of donors and samples enrolled into the analysis. Thus, the obtained results might be out of proportion to the data reported. How were the numbers of study samples determined, i.e. how was the study powered?

2.     Since RA female  patients received treatment with a TNf-a inhibitor and were also supplemented with 800 IU per day of vitamin D, a two-way ANOVA analysis of variances should be additionally performed to assess the joint effect of the two factors on the assessed bone metabolism parameters.

3.     Table 3 shows that the number of patients with the duration of RA of 1-5 years was 84 people, and the number of patients with the duration of the disease of 5-10 years - was 16 people. This data is not consistent with the information that the total number of RA patients enrolled in the trials was 50.

Author Response

REVIEW 1 REPORT

Comments

In the manuscript entitled: “Changes in bone metabolism in patients with rheumatoid arthritis during tumor necrosis factor inhibitor therapy” the Authors aimed to investigate the effect of the one year therapy with TNF-a inhibitors on the bone metabolism assessed by biochemical reference markers of bone synthesis (P1NP - procollagen type 1N-terminal propeptide)  and bone resorption  (b-CTX - beta crosslaps C-terminal telopeptide of collagen type I beta-crosslaps) and inorganic matrix markers (total and ionized calcium, phosphorus and alkaline phosphatase) as well as hormones closely related to bone metabolism such as parathyroid hormone (PTH) and vitamin D level.  The effectiveness of 12-month therapy with TNF inhibitor (etanercept, adalimumab, golimumab and infliximab) on bone metabolism improvements in 50 female patients with rheumatoid arthritis was assessed in this study. Authors conclude that yearlong application of TNF inhibitors has the positive effect on bone metabolism, as indicated by an increase in bone-forming markers. The undertaken research topic is of significant importance in clinical practice.

The manuscript, however, has some limitations, the most important of which are:

  1. A small number of donors and samples enrolled into the analysis. Thus, the obtained results might be out of proportion to the data reported. How were the numbers of study samples determined, i.e. how was the study powered?
  2. Since RA female patients received treatment with a TNf-a inhibitor and were also supplemented with 800 IU per day of vitamin D, a two-way ANOVA analysis of variances should be additionally performed to assess the joint effect of the two factors on the assessed bone metabolism parameters.
  3. Table 3 shows that the number of patients with the duration of RA of 1-5 years was 84 people, and the number of patients with the duration of the disease of 5-10 years - was 16 people. This data is not consistent with the information that the total number of RA patients enrolled in the trials was 50.

RESPONSES TO REVIEWER 1

First, we wish to take this opportunity to thank you for providing us with such valuable suggestions, as addressing your comments has certainly elevated the quality of our work. Please find our responses to each of the issues raised, which are addressed in the manuscript in relevant sections.

  1. A small number of donors and samples enrolled into the analysis. Thus, the obtained results might be out of proportion to the data reported. How were the numbers of study samples determined, i.e. how was the study powered?

Authors’ reply:Thank you for seeking clarification regarding our study design and sample selection criteria. In response, we wish to note that the research presented in the manuscript was designed as a 12-month-long retrospective/prospective study that would include all patients treated with a biological drug from the TNF-inhibitor group in the Special Hospital for Rheumatic Diseases in Novi Sad, Serbia.In order to obtain valid results, our sample was as homogeneous as possible in terms of disease type, gender, age, disease length and activity, and treatment method. Our cohort comprised of 50 female patients, which was sufficient, as 48 individuals was the minimum required to attain 10% margin of error and 85% confidence level.

  1. Since RA female patients received treatment with a TNf-a inhibitor and were also supplemented with 800 IU per day of vitamin D, a two-way ANOVA analysis of variances should be additionally performed to assess the joint effect of the two factors on the assessed bone metabolism parameters.

Authors’ reply: Thank you for making this valuable suggestion. Accordingly, we have now conducted a two-way ANOVA analysis of variances in order to examine the joint effect of TNF-α inhibitors and vitamin D on the examined bone metabolism parameters. The obtained results did not exceed the threshold of statistical significance (p = 0.037).

  1. Table 3 shows that the number of patients with the duration of RA of 1-5 years was 84 people, and the number of patients with the duration of the disease of 5-10 years - was 16 people. This data is not consistent with the information that the total number of RA patients enrolled in the trials was 50.

Authors’ reply: We apologize for this ambiguity. The figures the reviewer mentioned above (84 and 16) in fact refer to the percentage of the sample, so that they will correspond to 42 and 8 individuals, respectively, totaling 50 patients. These mistakes are now clarified in Table 1.

Reviewer 2 Report

This article investigated the influences of yearlong tumor necrosis factor inhibitor therapy on bone metabolism through analyzing bone mineral density and the biochemical markers. The manuscript gives a very good feedback from the clinic to the drug developer. Generally the manuscript is well written and organized. I would suggest that the manuscript can be accepted if the following comments are taken into account.

 1.       The authors should reorganize the Keywords to make it more representatively and accurately.

2.       Only female RA patients were selected for the studied. A brief explanation should be included in the Introduction.

3.       In the study sample, only one patient was administrated with Infliximab (Table 1). The authors later mentioned that “Since only 2% of patients received infliximab, its influence on the observed parameters was not assessed (lines 225-226)”. Is it better to exclude this sample from the whole article?

4.       Discussion is suggested to be simplified. The key findings should be highlighted.

Others

1.      Please use the spacing correctly. Eg. “TNF-αpromotes” (line 40), “joint count (SJC),as well as” (Line 130), “3.2- 5.1” (Line 136) ect.

2.       Please insert a hyphen between “anti” and “cyclic” (Page 4, line 188).

Author Response

REVIEW 2 REPORT

Comments and Suggestions for Authors

This article investigated the influences of yearlong tumor necrosis factor inhibitor therapy on bone metabolism through analyzing bone mineral density and the biochemical markers. The manuscript gives a very good feedback from the clinic to the drug developer. Generally the manuscript is well written and organized. I would suggest that the manuscript can be accepted if the following comments are taken into account.

  1. The authors should reorganize the Keywordsto make it more representatively and accurately.
  2. Only female RA patients were selected for the studied. A brief explanation should be included in the Introduction.
  3. In the study sample, only one patient was administrated with Infliximab (Table 1). The authors later mentioned that “Since only 2% of patients received infliximab, its influence on the observed parameters was not assessed (lines 225-226)”. Is it better to exclude this sample from the whole article?
  4. Discussion is suggested to be simplified. The key findings should be highlighted.

Others

  1. Please use the spacing correctly. Eg. “TNF-αpromotes” (line 40), “joint count (SJC),as well as” (Line 130), “3.2- 5.1” (Line 136) ect.
  2. Please insert a hyphen between “anti” and “cyclic” (Page 4, line 188).

RESPONSES TO REVIEWER 2

First, we wish to take this opportunity to express our gratitude for the time and effort you dedicated to such a detailed review, as well as apologize for the typos and other issues that you so kindly identified in the first version of our manuscript. Thank you for giving us the opportunity to clarify some misunderstandings and thereby improve our work.We have addressed all your suggestions in the main text, and have revised the Discussion section thoroughly, as explained below where we respond to each of your comments in turn.

  1. The authors should reorganize the Keywordsto make it more representatively and accurately.

Authors’ reply: Thank you for making this request. We have now presented the keywords in the following order: Rheumatoid arthritis; bone metabolism; bone mineral density; osteoporosis inhibitor tumor necrosis factor alpha

  1. Only female RA patients were selected for the studied. A brief explanation should be included in the Introduction.

Authors’ reply: Thank you for seeking clarification regarding the decision to solely focus on RA patients. As this was a one-year retrospective/prospective study conducted at the Special Hospital for Rheumatic Diseases in Novi Sad, Serbia, where majority of patients are female and have RA, and some are prescribed a biological drug from the TNF-inhibitor group, we felt that examining the impact of this treatment mode on our patient population was highly pertinent. When forming our sample, we made sure that it was as homogenous as possible with respect to disease type, gender, age, disease duration and activity, and treatment method.

  1. In the study sample, only one patient was administrated with Infliximab (Table 1). The authors later mentioned that “Since only 2% of patients received infliximab, its influence on the observed parameters was not assessed (lines 225-226)”. Is it better to exclude this sample from the whole article?

Authors’ reply: We really appreciate this question, as it indicated that the readers may also wonder about the inclusion of a single patient receiving Infliximab into our analyses. However, our primary goal was to determine the impact of all applied TNF inhibitors on the bone metabolism of RA patients. Thus, we felt that including data (albeit limited) pertaining to Infliximab was appropriate. On the other hand, as our secondary goal was to establish whether the TNF inhibitor type influenced the changes in the observed bone metabolism parameters, Infliximab was excluded from these analyses, given that it accounted for only 2% of the gathered data.

  1. Discussion is suggested to be simplified. The key findings should be highlighted.

Authors’ reply: Thank you for requesting that we revise and simplify the Discussion section. Accordingly, we have modified most of its contents, as you can see below, and we hope that this new version is more concise and informative. As we have also added a new reference (under citation number 31), we have renumbered all other sources both in the text and in the Reference section.

Discussion

In extant literature, presence of numerous pro-inflammatory cytokines in rheumatoid arthritis (RA) is associated with localized inflammatory bone resorption and generalized bone loss. Available findings further indicate that the receptor activator of nuclear factor kB (RANK)-RANK ligand (RANKL) system is the main driver of inflammatory bone resorption [30]. On the other hand, the use of TNF inhibitors has been shown to affect bone metabolism by increasing the P1NP and osteocalcin (OC) serum level (as bone synthesis markers), and decreasing the levels of serum b-CTX and RANKL (as bone resorption markers in RA), thus slowing down generalized osteoporosis and the development of periarticular erosions [31].

Guided by this evidence, the aim of the present study was to evaluate the impact of 12-month use of TNF inhibitors on changes in bone mineral density and biochemical markers of bone synthesis (P1NP) and resorption (b-CTX). These markers were chosen as our sample comprised solely of women aged 51.50±3.94 years, and it is widely known that estrogen affects bone homeostasis directly (via the effect on bone cells through the RANKL-RANK-OPG pathway), as well as indirectly—through an immune mechanism by mitigating the increase in cytokine (such as IL-1, IL-6 and TNF) production caused by estrogen deficiency. In this context, TNF-α (produced by T-lymphocytes in the bone marrow) is particularly significant, as it is the most influential cytokine in bone loss caused by estrogen deficiency. In our cohort, menopause duration did not exceed five years, which is relevant as estrogen levels decrease rapidly during menopause, which in turn activates the differentiation and proliferation of osteoclasts, while inhibiting the action of osteoblasts and increasing osteocyte apoptosis. All these effects accelerate bone resorption, as confirmed by extant evidence indicating that the onset of estrogen decline coincides with a phase of rapid bone loss, resulting in a 10% and 5% bone mass loss at the spine and hip, respectively, during the first five years of menopause. Some estimates further suggest that after 10−15 years of menopause, about 50% of trabecular and 30% of cortical bone is lost [32]. Our findings indicate that 12-month use of TNF inhibitors attenuates the decline in BMD measured at the L1−L4 level as well as in the femoral neck area. Moreover, although the increase in BMD values for the lumbar spine and hip did not exceed the threshold of statistical significance, we observed statistically significant improvements in the corresponding T-score (SD) values. Our findings are in agreement with the results reported by Nutz et al. based on a meta-analysis of 15 studies that included 12-month TNF-inhibitor treatment in patients with RA [33].

Similarly, Zerbini and colleagues analyzed 28 studies in which participants received TNF inhibitors and observed their beneficial effect on preserving or increasing BMD in the lumbar spine and hip, as well as achieving a better biochemical bone marker profile [34]. Findings reported by other authors yielded similar conclusions [35−38]. It is also worth noting that, by analyzing the effect of TNF inhibitors on bone metabolism in RA patients over a 15-month period, Jura-Poltorak et al. confirmed that it is effective in arresting bone loss. However, in their cohort, the improvements in BMD in the lumbar spine and femoral neck, as well as the corresponding T scores, failed to reach statistical significance. On the other hand, as the authors observed a statistically significant change in the levels of bone synthesis/resorption biochemical markers, it appears that these markers respond more rapidly to TNF inhibitor therapy compared to bone mineral density [39].

In the present study, a higher degree of bone resorption was associated with greater disease activity and b-CTX levels, which correlated with lower BMD values at the onset of TNF inhibitor therapy.Therefore, by monitoring b-CTX levels, we could predict further BMD loss.Our analyses also revealed that TNF inhibitors increased serum P1NP levels and mitigated the b-CTX increase, thus elevating the P1NP/b-CTX ratio. Moreover, TNF inhibitor use was positively correlated with changes in the bone biochemical marker levels, whichindicates a statistically significant increase in P1NP levels, as well as improvements in b-CTX (which did not exceed the threshold of statistical significance). Finally, the observed changes in P1NP and b-CTX upon completion of 12-month TNF inhibitor administration were associated with RA disease activity reduction.Similar findings were reported by Szulc and colleagues based on an investigation involving 54 patients treated with TNF inhibitors for 12 months [40].However, other authors reported statistically significant improvements in both parameters, while some also noted BMD increase in their cohorts [39].

To situate our results in this context, we segregated our sample into younger (below 50 years) and older (50+ years) groups and examined their outcomes. These comparisons revealed that greater improvements in P1NP and b-CTX were associated with younger age (30% and 10.5% for those under 50 vs. 26.8% and 6.6% for women over 50). Likewise, when the cohort was segregated by BMI, we determined that being underweight (BMI < 19), as well as overweight or obese (BMI > 25), was a risk factor for osteoporosis, as it resulted in more pronounced changes in bone metabolism. These observations concur with the available data, indicating that BMD declines more rapidly in obese individuals due to inadequate physical activity, hypertension, and suboptimal vitamin D levels [41]. In addition, adipose tissue secretes cytokines that affect bone tissue by increasing bone resorption, while adipokines affect the central nervous system by altering the influence of the sympathetic nervous system on bone tissue [42].Therefore, it is not surprising that the greatest changes in P1NP values (38%) and b-CTX (8.4%) were recorded in patients of normal weight.

Hypovitaminosis D is highly prevalent in patients suffering from inflammatory rheumatic diseases, especially RA, and may exacerbate the negative impact of inflammation on BMD. Vitamin D in its active form—25(OH)D—exhibits an immunoregulatory effect, which manifests through the regulation of monocytes and macrophages, as well as the activity of B and T cells. It is thus believed that vitamin D intake can reduce the production of pro-inflammatory cytokines such as TNF-α and IL-6, which play a key role in bone resorption in RA patients. Consequently, the potential beneficial role of vitamin D as a modulator of inflammation in RA is increasingly being discussed [43].

In our cohort, 12-month TNF inhibitor therapy resulted in a statistically significant reduction in disease activity (as measured by composite indices DAS28 SE and DAS28 CRP), while increasing average vitamin D, total calcium, ionized calcium, and phosphorus values. These observations are supported by the findings reported by Oelzner et al., indicating that high RA activity is associated with changes in vitamin D metabolism and increased bone resorption [44], which is expected given that lower serum vitamin D levels may contribute to negative calcium balance and inhibition of bone formation. Similar conclusions were reached by Scott et al. and Lin et al. based on their meta-analyses of available studies [45,46]. However, given the paucity of comparative studies focusing on the efficacy of different TNF inhibitors (mostly tocilizumab and abatacept), our findings regarding the changes in BMD and bone synthesis/resorption biochemical markers can only be directly compared with those reported by Sainaghi and colleagues with which they concur [47]. It is also worth noting that, even though 12-month administration of all tested biological drugs resulted in an increase in P1NP and b-CTX, the highest percentage increase in P1NP was achieved with adalimumab (34%), followed by golimumab (32.6%) and finally etanercept (18.7%). In contrast, b-CTX was most significantly modified by etanercept (9.21%), followed by adalimumab (7.48%) and finally by golimumab (4.94%).

The significance of our study derives from the inclusion of biologically naïve patients, who were selected based on clearly defined criteria, excluding individuals taking glucocorticoids and other drugs, as well as those suffering from diseases that are known to impact bone remodeling (which would affect any changes in BMD that occur during the study period). This rigorous research design allowed us to obtain a clearer picture of the effect of TNF inhibitors on BMD. Furthermore, several variables—SE, CRP, PTH, vitamin D 25(OH) level, DAS28 SE and DAS28 CRP, and self-reported quality of life—that could potentially affect bone metabolism were monitored. Based on these analyses, we hypothesize that TNF inhibitors are not only effective in controlling inflammation, but may also directly inhibit osteoclast activity.

The following reference was newly added:

  1. Hamar, A.; Szekanecz, Z.; Pusztai, A. et al. Effects of one-year tofacitinib therapy on bone metabolism in rheumatoid arthritis. Osteoporos Int 2021, 32, 1621–1629 . https://doi.org/10.1007/s00198-021-05871-0

  1. Please use the spacing correctly. Eg. “TNF-αpromotes” (line 40), “joint count (SJC),as well as” (Line 130), “3.2- 5.1” (Line 136) ec

Authors’ reply: Thank you for pointing out these typos in our original manuscript. We sincerely apologize for not reading it more thoroughly before the initial submission to the journal. All noted issues have now been rectified in the manuscript.

  1. Please insert a hyphen between “anti” and “cyclic” (Page 4, line 188).

Authors’ reply: Once again, we apologize for making these mistakes which have now been eliminated.

Reviewer 3 Report

1. At introdution section, please mention that TNF-alpha, and following NF-kappa B signaling regulate BMP signaling in osteoblast linage cells.  

2. Please mention NF-kappa B signaling activated by TNF-alpha in some detail.

Author Response

RESPONSES TO REVIEWER 3

All authors wish to thank you for the time and effort you have dedicated to the review of our manuscript and for providing us with constructive suggestions aimed at elevating its quality. In line with your highly detailed feedback, in the Introduction section, we have now explained NF-kappa B signaling in more detail (as shown below) and have added three new sources to the reference list.

Introduction

Rheumatoid arthritis (RA) is a systemic, autoimmune, chronic inflammatory disease that affects bone metabolism by increasing bone resorption, leading to osteoporosis and a high risk of fracture [1,2]. Presence of numerous pro-inflammatory cytokines in RA is associated with the activation of osteoclastogenesis, with tumor necrosis factor alpha (TNF-α) playing a leading role in this process. TNF-α stimulates the expression of the receptor activator of nuclear factor kB ligand (NF-kB), RANKL, which is instrumental in osteoclast differentiation and maturation. It can also exert its influence through the soluble receptor osteoprotegerin (OPG), or via stromal cells of the bone marrow lineage of osteoblasts, as well as by directly activating the cells in osteocalcate lineage [3]. The interaction between RANKL and its receptor-activator of nuclear factor kB (RANK) occurs on the surface of osteoclasts by forming a RANKL/RANK bond that leads to accelerated differentiation, maturation, activation and life extension of osteoclasts, which initiates the bone resorption process. The OPG soluble part of the receptor for RANKL, which is structurally homologous to RANK, by binding to RANKL forms a RANKL/OPG bond that inhibits the final phase of osteoclast differentiation as well as activation of matrix osteoclast suppression, while accelerating osteoclast apoptosis. Therefore, balance between RANKL and OPG is the main regulatory factor of the biological balance involving bone formation and resorption [4−6]. TNF-α can stimulate osteoclast precursors directly through TNF receptor 1 (TNFR1) signaling, while soluble TNF is responsible for mobilization of osteoclasts from the bone marrow. Binding of TNF-α to receptors activates intracellular cascades that include NF-κB and mitogenic activation of protein kinase, thus transmitting information from the receptor to the nucleus [7]. Osteoblast differentiation and activity is influenced by TNF-α while NF-kB signaling inhibits the regulation of bone morphogenetic protein (BMP) [8]. As Wnt uses a co-receptor (frizzled receptor and lipoprotein receptor 5 − LPR5) to activate, its transduction pathway is presently considered the most important for the control of osteoblast differentiation. In addition, in interaction with the Wnt signaling pathway, TNF-α plays an important role in the control and differentiation of osteoblasts, as TNF-α stimulates Dkk-1 (Dickkopf-related protein 1) expression of the endogenous Wnt signal inhibitor. During this process, Dkk-1 binds to the LDL-receptor LRP5 or LRP6 of osteoblasts, which inhibits their activity and enhances osteoclastogenesis [9]. By stimulating the production of sclerostin—a hormone produced by osteocytes—TNF reduces preosteoblast differentiation into osteoblasts and thus inhibits bone tissue formation [10,11].

3.Walsh, N.C.; Crotti, T.N.; Goldring S.R.; Gravallese, E.M. Rheumatic diseases: The effects of inflammation on bone. Immunol. Rev. 2005;208:228–251. doi: 10.1111/j.0105-2896.2005.00338.x.

7.Wysocki, T. Tumor Necrosis Factor in Rheumatoid Arthritis. Encyclopedia. Available online: https://encyclopedia.pub/entry/20537 (accessed on 16 February 2023).

8.Yi, S.-J.; Lee, H.; Lee, J.; Lee, K.; Kim, J.; Kim, Y.; Park, J.-I.; Kim, K. Bone Remodeling: Histone Modifications as Fate Determinants of Bone Cell Differentiation. Int. J. Mol. Sci. 201920, 3147. https://doi.org/10.3390/ijms20133147

Round 2

Reviewer 1 Report

The Authors correctly responded to all my comment described in the review report